# Evidence for Beneficial Physiological Responses of the Land Snail *Cornu aspersum* to Probiotics’ (*Lactobacillus plantarum*) Dietary Intervention

**DOI:** 10.3390/ani14060857

**Published:** 2024-03-11

**Authors:** Efstratios Efstratiou, Konstantinos Feidantsis, Vasiliki Makri, Alexandra Staikou, Ioannis A. Giantsis

**Affiliations:** 1Faculty of Agricultural Sciences, University of Western Macedonia, GR-53100 Florina, Greece; str.efstratiou@gmail.com (E.E.); makrivasil@bio.auth.gr (V.M.); 2Department of Fisheries and Aquaculture, University of Patras, GR-26504 Mesolonghi, Greece; kfeidant@upatras.gr; 3Department of Zoology, School of Biology, Aristotle University of Thessaloniki, GR-54124 Thessaloniki, Greece

**Keywords:** land snails, *Cornu aspersum*, probiotic, *Lactobacillus plantarum*, physiological responses, apoptosis, heat shock proteins, mitochondrial complex I (ND2)

## Abstract

**Simple Summary:**

Nutrition is one of the most important physiological processes for the growth and welfare of farmed animals, while interactions between the intestinal microflora and the host systems provide a key mechanism for enhancing and maintaining homeostasis. Probiotics exert a positive impact on organisms’ growth and immune response, so the administration of rations enriched with probiotics is recommended. Although the physiological role of probiotics on vertebrates’ growth and stress responses has been extensively studied in vertebrates, little is known regarding their effects on economically important invertebrates such as land snails. The aim of the present study was to investigate the effects of probiotic-enriched rations on the physiological responses in several tissues of the economically important farmed terrestrial snail, *Cornu aspersum*. Despite the absence of a direct effect on growth rate, an intense physiological response by various biomarkers was observed in all examined tissues.

**Abstract:**

A large variety of microorganisms ingested with food constitute animals’ intestinal microflora, enhancing and maintaining the homeostasis of the host. Rations enriched with probiotics are a method recommended to prevent undesirable conditions. To date, research has been limited to farmed animals and reared fish, creating a knowledge gap concerning the effect of probiotics on the growth rate, physiological responses, and energy metabolism of invertebrates such as the land snail *Cornu aspersum*. Herein, juvenile snails (26.23 ± 0.5 mm shell diameter and 8.23 ± 1.02 g body weight) were fed *L. plantarum* probiotic-enriched rations in two different proportions (1.25 mg and 2.5 mg), and their growth rate was monitored for three months. Additionally, the RNA/DNA and Bax/Bcl-2 ratios, HSP gene expression and protein levels, and ND2 expression, were measured in the hepatopancreas, digestive tract, and mantle. Although the snails’ growth rate was not affected, the RNA/DNA ratio presented an increase in various tissues, indicating an intense physiological response. Also, probiotic administration demonstrated low levels of the Bax/Bcl-2 ratio. HSP levels were higher in the presence of probiotics, probably signaling an attempt by the animal to face potentially stressful situations. Finally, ND2 expression levels in the hepatopancreas indicate intense metabolic and antioxidant activity.

## 1. Introduction

Although snail farming represents an innovative entrepreneurial activity leading to the yield of a valuable food product of export orientation for many countries, the insufficient knowledge of producers about the pulmonated snail’s biology, ethology, and productive farming processes poses several difficulties [1]. The use of modern techniques, the effective management of the production cycle, as well as integrated zootechnical support can ensure high yields and positive financial results for the snail farming industry [1,2]. In addition to the type of facilities and the safety conditions, an important parameter for snail farming is nutrition. In land snail farming, the employed type of ration plays a significant role in the animals’ welfare, aiming at rapid growth and enhanced viability as well as minimum nutritional costs and environmental impacts [3,4,5].

Rations enriched with probiotics have recently started to be utilized in animal farming and various strains of probiotics are commercially available [6,7]. They are widely used in animal husbandry and fish aquaculture, since it has been demonstrated that they improve growth and feed conversion, and the overall health status of animals [8,9,10,11,12,13]. Strains of the genus *Lactobacillus* have been used especially as probiotics for several years, proving their safe consumption [7,14].

The exact cellular mechanisms related to the beneficial effects of probiotics are not well known. Several investigations, however, have shown that these mechanisms might be associated with stress responses, including several members of heat shock proteins (HSPs), and/or increased efficiency of antioxidant and autophagy cellular mechanisms, resulting in higher cell viability [15]. It is well known that HSPs are involved in several physiological processes including growth, lifespan, and reproduction stress tolerance [16,17,18]. During recent years, dietary approaches have been suggested as a promising way of increasing the antioxidant defense in several organisms such as different fish and shellfish species [19]. On the other hand, probiotics enhance disease resistance, decrease stress susceptibility, and improve general vigor [20,21]. Moreover, properties of *Lactobacillus* bacteria, such as S-layer proteins and exopolysaccharides, which increase hydrophobicity and hydrophilicity, respectively, are involved in the formation of intestinal biofilm [22,23]. These properties also enhance the antimicrobial and immunomodulatory effects by the inhibition of TNFα production [23,24], making this strain a promising tool in animal feeding.

However, the beneficial effects of *Lactobacillus* genus on growth and stress responses of farmed animals were studied mainly on several vertebrate species including fish, while little is known regarding the nutritional effects of *Lactobacillus* in invertebrate farming systems. It has been recently reported that isolated strains of the genera *Lactobacillus* from the intestinal tract of the terrestrial gastropods *Cornu aspersum* (O.F. Müller, 1774) act as probiotics by enhancing their immune system when administered together with the diet [25,26]. A total of 48 candidate probiotic strains of *Lactobacillus* and 15 potential pathogen strains of *Listeria monocytogenes* were isolated and investigated at a functional, phenotypic, and genetic level [25,26]. The aim of the present work was to further investigate whether the use of probiotics exerts beneficial physiological effects associated with protein synthesis, mitochondrial metabolism, apoptosis, and stress responses on immature individuals of the land snail species *C. aspersum*. It is well known that the ratio of RNA/DNA is closely related to protein synthesis, and it is a well-established ecophysiological indicator of organisms’ activity [27,28,29]. Moreover, it provides information on growth, secretion, and reproduction under various environmental conditions [27,28], contributing to the assessment of the metabolic state and physiology [29]. Since growth is closely related to ingested food and energy allocation to different tissues, we decided to study the effects of probiotics on the RNA/DNA ratio in three different tissues as digestive tract, hepatopancreas, and mantle in immature individuals of species *C. aspersum*. Additionally, we determined the relative expression of mitochondrial complex I (NADH dehydrogenase—ND2) which is a strong indicator of respiratory chain and mitochondrial oxidation related to aerobic metabolism. Additionally, we studied the expression of the HSP70 and HSP90 genes, which, apart from stress responses, are related to the folding and aggregation of de novo protein synthesis. Especially, HSP90 and its co-chaperones orchestrate crucial physiological processes such as cell survival, cell-cycle control, hormone signaling, and apoptosis [30]. Moreover, we studied the ratio Bax/Bcl-2 as an indicator of the activation or not of apoptotic processes and the cells’ survival. The balance between the pro-apoptotic Bax and the anti-apoptotic Bcl-2 can determine the cell fate and consequently the ratio Bax/Bcl-2 indicates the triggering of pro apoptotic processes [31].

## 2. Materials and Methods

### 2.1. Animal Maintenance

#### 2.1.1. Snail Collection

A total of 180 farmed *C. aspersum* individuals, with an average shell diameter (D) of 26.23 ± 0.5 mm and body weight of 8.23 ± 1.02 g, were divided into equal groups and placed into glass boxes (20 × 20 × 40 cm). The temperature of the room was adjusted to 22 ± 2 °C, while humidity was maintained at high levels (70–90%) by placing a wet sponge inside the boxes. The experiment ran under a photoperiod of L:D ratio 12:12 h. The snails were acclimated for 1 week under the above-mentioned conditions without feeding. All individuals were sexually immature to avoid mating throughout the experiment. Their growth was monitored by measuring the shell diameter (D) every 30 days. Dead individuals were replaced by snails of the same size to keep the population density within each container constant. This replacement ensured that any difference found in growth rate of snails among rearing boxes would not be attributed to density effects. Moreover, the newly added snails were not taken into account for tissue sampling. The experimental process lasted three months (December 2022–February 2023).

#### 2.1.2. Snail Feeding

After the period of acclimation, the snails were fed with commercial rations (Zoothreptiki Animal Nutrition SA—Sapounas Bros SA cat number 23A) (Table 1). According to the experimental procedure, the feed was divided into three groups. The first one was used without the addition of probiotics. The second and the third groups were enriched with 1.25 mg and 2.5 mg of the species *L. plantarum* per 5 g of rations, respectively. These dosages were selected based on previous investigations of probiotic administration of bacterial taxa in various snails and other farmed animals [23,25,26,32]. More specifically, capsules of powder with probiotic strain *L. plantarum* from the Swanson company (product code: SWV-19016) were administered. Each capsule contained 25 mg with 10 billion CFU/g (colony forming units) when manufactured, providing an effective level of bacteria. Thereafter, equal quantity from all three groups of feed was placed into petri dishes of the same size and each petri was placed correspondingly into the glass boxes. Snails were able to feed from the beginning until the last day of the experimental conditions. The group of individuals fed rations without the addition of probiotic were characterized as the control, while the second and the third groups fed with a mixture containing 1.25 mg and 2.5 mg were characterized as the FC and 2FC groups, respectively. Four glass boxes (replicates) were used for each treatment. From each glass box, five individuals were pooled and used for analyses. Snails kept under the above described feeding conditions for three months.

#### 2.1.3. Tissue Sampling

A total of 3 samplings were carried out every 30 days. During snail sampling, individuals were drawn out of the boxes and were immediately placed into liquid nitrogen. For tissue sampling, individuals were thawed, and the shell was removed to reveal the visceral mass. Thereafter, the hepatopancreas and digestive tract (without the buccal mass) were excised and were immediately placed in Eppendorf tubes and stored at −80 °C for further analyses. All further analyses were conducted in pools per five individuals.

### 2.2. Analytical Procedures

#### 2.2.1. RNA/DNA Ratio

Extraction of total RNA and genomic DNA was performed according to the AllPrep DNA/RNA Micro Kit (cat. No 80284, QIAGEN, Hilden, Germany). All solutions were prepared according to the manufacturer’s instructions. Extracted RNA was stored at −80 °C and DNA at −25 °C. DNA/RNA ratio was calculated based on concentrations of extracted RNA and DNA from each sample, whereas A_260/230_ and A_260/280_ values were calculated in a Q5000 spectrophotometer (Quawell, Beijing, China). DNA and RNA integrity were validated in an agarose gel after electrophoresis stained with ethidium bromide.

#### 2.2.2. Complementary DNA Synthesis

Reverse transcription was performed according to “PrimeScript™ 1st strand cDNA Synthesis Kit” (TaKaRa). The mixture was prepared according to the manufacturer’s instructions and incubated under appropriate conditions in a thermocycler for (a) the process of reverse transcription (37 °C, 15 min), (b) the reverse transcriptase inactivation (85 °C, 5 s), and (c) the incubation at 4 °C. The product was stored in a freezer at −25 °C.

#### 2.2.3. Real-Time Polymerase Chain Reaction

PCR primer sequences for the *C. aspersum* ND2 gene were designed based on the whole mitochondrial genome of *C. aspersum*, while, for the HSP70 gene, they were selected according to Bouétard et al. [33] as presented in Table 2. The efficiency of both of them was greater than 95% (Table 2). The 18s rRNA gene, which has been used in other studies of snails as a reference gene for the expression normalization of target genes [33], was used as a housekeeping gene to normalize target gene transcript levels. The 2^−ΔΔCt^ method was used to estimate messenger RNA (mRNA) abundance [34]. Relative gene expression levels were normalized by eukaryotic reference gene r18s.

#### 2.2.4. Immunoblotting/SDS-PAGE

The preparation of samples for SDS-PAGE and immunoblot analysis are based on well-established protocols. Specifically, 50 mg of frozen tissue were immediately homogenized in 3 mL g^−1^ of cold lysis buffer (20 mM β-glycerophosphate, 50 mM NaF, 2 mM EDTA, 20 mM Hepes, 0.2 mM Na_3_VO_4_, 10 mM benzamidine, pH 7, 200 μM leupeptin, 10 μΜ trans-epoxy succinyl-Lleucylamido-(4-guanidino) butane, 5 mM dithiotheitol, 300 μΜ phenyl methyl-sulfonyl fluoride (PMSF), 50 μg ml^−1^ pepstatin, 1% *v*/*v* Triton X-100), and extracted on ice for 30 min. Samples were centrifuged (10,000× *g*, 10 min, 4 °C) and the supernatant was boiled with SDS/PAGE sample buffer (330 mM Tris-HCl, 13% *v*/*v* glycerol, 133 mM DTT, 10% *w*/*v* SDS, 0.2% *w*/*v* bromophenol blue) in a 3:1 ratio (40 μL buffer for 120 μL supernatant). Equivalent amounts of protein (50 μg) were separated on 10% and 0.275% (*w*/*v*) acrylamide and bisacrylamide gels using a Bio-Rad Mini-PROTEAN Tetra Vertical Electrophoresis Cell (Bio-Rad, Hercules, CA, USA). When the protein front was extracted from the gels, the gels were placed into a Bio-Rad Trans-Blot SD Semi-Dry Transfer Cell (Bio-Rad, Hercules, CA, USA) and the protein content of the gels was electrophoretically transferred for 60 min onto nitrocellulose membranes (0.45 μm, Schleicher and Schuell, Keene, NH 03431, USA) presoaked with transfer buffer (25 mM Tris, 192 mM glycine, 20% *v*/*v* methanol, pH 8.3).

Non-specific binding sites on the membranes were blocked with 5% (*w*/*v*) non-fat milk in TBST (20 mM Tris-HCl, pH 7.5, 137 mM NaCl, 0.1% (*v*/*v*) Tween 20) for 30 min at room temperature. The resulting nitrocellulose membranes were subjected to overnight incubation with: polyclonal rabbit anti-bcl2 (2872, Cell Signaling, Beverly, MA, USA), polyclonal rabbit anti-bax (2772, Cell Signaling, Beverly, MA, USA), monoclonal mouse anti-HSP70 (H5147, Sigma, Darmstadt, Germany), and monoclonal mouse anti-HSP90 (H1775, Sigma, Darmstadt, Germany). Quality transfer was assured by Ponceau stain and actin (anti-β actin 3700, Cell Signaling, Beverly, MA, USA). Antibodies were diluted as recommended by the manufacturer’s guidelines. After washing in TBST (3 periods, 5 min each time), the blots were incubated for 60 min at room temperature with horseradish peroxidase-linked secondary antibodies (anti-mouse HRP-linked antibody (7076, Cell Signaling, Beverly, MA, USA), anti-rabbit HRP-linked antibody (7074, Cell Signaling, Beverly, MA, USA)), washed again in TBST (3 periods, 5 min each time), and the bands were detected using enhanced chemiluminescence (Signal Fire ECL Reagent, 6883, Cell Signaling, Beverly, MA, USA) with exposure to Fuji Medical X-ray films. Films were quantified by laser-scanning densitometry (GelPro Analyzer Software version 3, GraphPad). Each of the examined proteins was normalized with its respective β-actin and results are expressed as relative protein levels.

We have to state that due to the high levels of mucus in the mantle, no reliable results regarding Western-blot analysis in this tissue were obtained and therefore are not presented.

### 2.3. Statistical Analysis

Changes regarding relative biochemical indicators were tested by one-way analysis of variance (ANOVA) (GraphPadInstat 3.0) for significance at the 5% level (*p* < 0.05) between examined groups. Moreover, statistically significant differences were tested by two-way (GraphPad Prism 5.0) analysis of variance (ANOVA). Sampling time points and probiotic treatments (FC and 2FC) were regarded as fixed factors. Post-hoc comparisons were performed by the employment of the Bonferroni test. Values are presented as means ± standard deviation (S.D.). Due to the small sample size (n = 4), homogeneity of variance was not tested, since normality tests have little power to conclude whether or not a small sample of data comes from a Gaussian distribution.

## 3. Results

### 3.1. Snail Growth and Mortality

Throughout the experimental period, the snails grew very little in diameter (D) and, additionally, no difference in growth was observed among the three groups of snails fed with different rations. Furthermore, no difference was observed in percentage mortality among the three groups (Table 3).

### 3.2. RNA/DNA Ratio

The changes in the RNA/DNA ratio in the hepatopancreas, digestive tract, and mantle are depicted in Figure 1. As shown, a significant increase in the RNA/DNA ratio was observed in all examined tissues of the control group within the first 30 days, which was more potent, however, in the hepatopancreas and digestive tract. In the hepatopancreas, after a drop of control levels by day 60, the value of the RNA/DNA ratio exhibited a slight increase by day 90. Similarly, the presence of probiotics at the FC and 2FC groups caused a significant increase in the RNA/DNA ratio within the first 30 days, which, however, was comparatively less than that of the control group. Furthermore, the presence of 2.5 mg of probiotics (2FC group) seems to be less effective than that of 1.25 mg (FC group). Similar to the control group, the values of the RNA/DNA ratio decreased by day 60 and 90 in both groups treated with the probiotics. However, the statistical analysis revealed that the RNA/DNA ratio remained at higher levels compared to those of the control on day 0.

Similar to the hepatopancreas, probiotics caused a significant increase in the RNA/DNA ratio in the digestive tract within the first 30 days. Compared to the hepatopancreas, however, there was no statistical difference between the control and FC groups. On the other hand, although the ratio of RNA/DNA remained higher in the 2FC group than that on day 0, its value was determined to be significantly lower than that of the control and FC groups. Day 60 was characterized by an even drop in the RNA/DNA ratio in the control and FC groups and a further decrease in the 2FC group. Compared to day 60, no further decrease in the RNA/DNA ratio was observed in the digestive tract of the control group on day 90. On the contrary, this day was characterized by a two-fold increase in the RNA/DNA ratio in both the FC and 2FC groups.

The mantle exhibited a gradual increase in the RNA/DNA ratio in the control group by day 60 and remained at high levels by day 90. A similar pattern of increase was observed for the FC and 2FC groups as well, but to a lesser degree than those observed for the control group. The main effects of treatment and exposure time, as well as the factor interactions were significant (*p* < 0.0001).

### 3.3. Expression of the ND2 Dehydrogenase Gene

The pattern of changes in the ND2 mRNA levels in the hepatopancreas, digestive tract, and mantle are shown in Figure 2. In the hepatopancreas, the ND2 mRNA levels increased significantly on all sampling days, exhibiting a marked increase on day 90. No significant changes were observed in the FC and 2FC groups within the first 30 days, while both groups exhibited marked increases in the ND2 mRNA levels on days 60 and 90. Specifically, the ND2 mRNA levels increased by 30 and 20 folds during the days 60 and 90, respectively. The 2FC group presented a gradual increase in the ND2 mRNA levels, which were increased significantly on day 90. It should be pointed out, however, that the increases in the expression of the ND2 gene remained at lower levels than the corresponding level of the control group on day 90.

In the digestive tract, the relative expression of the ND2 gene presented a similar increase in the control and FC groups within the first 30 days. Comparatively, however, the 2FC group exhibited about a 40-fold increase compared to day 0. The levels of ND2 mRNA, after a transient decrease by day 60, increased thereafter in the control group by day 90. Compared to day 30, the levels of ND2 mRNA increased further in the FC group by day 60, followed by a transient decrease by day 90. Similar to the control group, the levels of ND2 mRNA were fluctuated in the 2FC group but remained at higher levels than those of the control group on day 0.

In the mantle, significant statistical increases in the relative expression of the ND2 gene were determined in the control group on days 30 and 60. In contrast to the FC group, however, such an increase was observed only in the 2FC group, shown by a marked increase on day 60. On days 30 and 60, both the control and the FC group exhibited a marked increase in the levels of ND2 mRNA on day 90, which comparatively were two-fold higher in the control group than in the FC group. On the contrary, no significant increase in the ND2 mRNA levels was observed in the 2FC group by day 90. The main effects of treatment and exposure time, as well as the factor interactions were significant (*p* < 0.0001).

### 3.4. HSP70 Relative Gene Expression

The relative expression of the HSP70 gene in the hepatopancreas, digestive gland, and mantle is depicted in Figure 3. In hepatopancreas, all groups exhibited significant increases in the relative expression of the HSP70 gene. Comparatively, however, the presence of probiotics seems to cause a faster and stronger relative gene expression within the first 30 days of growth. It should be pointed out, however, that the FC group presented a gradual decrease from day 30 to day 90.

In the digestive tract, marked increases in the HSP70 mRNA levels were observed in the control group mainly on days 30 and 60. The FC group presented a gradual increase in the relative expression of the HSP70 gene with the relative mRNA levels to significantly peak on day 90. Significant increases, but at lower levels, in the relative expression of the HSP70 gene were observed in the 2FC group mainly on days 30 and 60.

In the mantle, the presence of probiotics caused statistically significant increases in the HSP70 mRNA levels in the control group on days 30 and 60, but a marked increase on day 90. On the contrary, an increase in the HSP70 mRNA levels was determined in the FC group only on day 90. The main effects of treatment and exposure time, as well as the factor interactions were significant (*p* < 0.0001).

### 3.5. HSP70 and HSP90 Levels

Relative levels of HSP70 and HSP90 (HSP70/β-actin and HSP90/β-actin levels) in the hepatopancreas and digestive tract are exhibited in Figure 4 and Figure 5, respectively. In general, in both tissues, relative HSP70 levels were significantly increased in both the FC and 2FC groups compared to the control group throughout the experimental period of 90 days. The 2FC group exhibited significantly higher values compared to the FC group on days 60 and 90, while such a difference was not observed on day 30 (Figure 4). The main effects of treatment and exposure time, as well as the factor interactions were significant (*p* < 0.0001).

A similar pattern of changes was observed regarding HSP90 relative levels in both examined tissues. Specifically, both FC and 2FC groups exhibited significantly increased relative HSP90 levels compared to the control group throughout the 90-day duration of the experiment. The 2FC group exhibited significantly higher values compared to the FC group on days 60 and 90, while on day 30, the 2FC group exhibited significantly lower relative HSP90 levels in both tissues compared to the FC group (Figure 5). The main effects of treatment and exposure time, as well as the factor interactions were significant (*p* < 0.0001).

### 3.6. Bax/Bcl-2

The changes in relative Bax levels (Bax/β-actin levels), relative Bcl-2 levels (Bcl-2/β-actin levels) and the ratio Bax/Bcl-2 in the hepatopancreas and digestive tract are depicted in Figure 6 and Figure 7, respectively. In both examined tissues, the relative Bax levels were significantly decreased in both the FC and 2FC groups compared to the control group, with the 2FC group representing even significantly lower levels in relation to the FC group. Relative Bcl-2 levels remained the same throughout the experiment. Correspondingly, the Bax/Bcl-2 levels in the hepatopancreas revealed a pattern similar to that of Bax, with levels significantly decreased in both the FC and 2FC groups compared to the control group, while the 2FC group represented even significantly lower levels in relation to the FC group. The main effects of treatment and exposure time, as well as the factor interactions were significant (*p* < 0.0001).

## 4. Discussion

There is an increased worldwide interest in the utilization of probiotics to improve health parameters in animals and humans [35]. In addition, the enrichment of diets with probiotic strains is a common practice in production animal breeding systems to improve their welfare and productivity [36,37,38,39,40,41,42]. Although research has been carried out on the use of probiotics to enhance snail immune defense as well as on the microbiological content of the gastrointestinal tract [25,26,43], so far, the potential beneficial effects of lactic acid bacteria (in the host’s diet) on the biochemistry and physiology of gastropods of economic interest have not been evaluated. Keeping this in mind, the present study aimed to explore the response of the terrestrial snail *C. aspersum*, fed rations enriched with probiotics, by means of gene expression in both mRNA and protein levels. Since our study constitutes the first attempt to assess the biochemical response of a land snail in probiotics, only a few representative general biomarkers were examined. It should be noted that this species constitutes one of the main species of farmed land snail in the Greek area [1]. Farmed snails are vulnerable to pathogenic bacteria which can affect their viability, as well as endanger the health of consumers [44]. Due to its significant economic value, this edible snail usually becomes the principal representative among gastropods at research level.

It is well known that the amount of DNA in cells remains constant and it can therefore be used as an indicator of cell number or biomass, while the RNA/DNA ratio provides a good indication of the rate of the protein synthesis during organisms’ growth [45]. However, a quick change in mRNAs levels in the tissues under several conditions may influence the RNA/DNA ratio. The data obtained in the present work indicate that the RNA/DNA ratio is timely and tissue dependent in the snails. Bulow [46] supported that at an RNA/DNA ratio less than two, the body mass of the fish *Notemigonus crysoleucas* (Mitchill, 1814) is decreased, while at ratios higher than four, the body mass exhibits a rapid increase. Similar inferences were reported by Kim et al. [47], who observed that the larvae of the species *Takifugu obscurus* (Abe, 1949) maintained the minimum growth rate when the RNA/DNA ratio exceeded the limit of two. In the present study, the RNA/DNA ratio in all three tissues and three dietary groups was below 1.5 except for the value of 3.5 for the RNA/DNA ratio in the digestive tracts of the FC group on day 90. Nevertheless, in the mantle of the FC group, the RNA/DNA does not represent a strong indication that snail body weight will begin to increase. As mentioned in the Section 2, one week of starvation preceded the period of feeding. Consequently, the quick increase in the RNA/DNA ratio in the hepatopancreas of the control group within the first 30 days might indicate initially an increased demand for energy turnover, which is decreased during long term feeding. Such an increased energy demand perquisite increased de novo synthesis of functional and structural proteins. On the other hand, the lower RNA/DNA ratio determined at the FC and 2FC groups might be due to a higher energy absorption from the food assimilation in the digestive tract in the presence of probiotics [48]. The latter is consistent with the positive impact of probiotics in the digestive tract of the FC and 2FC groups, as it is reflected by the increased RNA/DNA ratio compared to the control group. Snails fed the probiotic reached their maximum growth rate in 2–3 months, having a continuous growth trend up to 150–170 days, in contrast to previous research, in which individuals fed with unenriched rations reached their maximum size in 110–130 days [25]. Because these differences are species- and probiotic-specific, future research should be focused on other beneficial microorganisms and other terrestrial invertebrates as well.

The significant gene expression of complex I (ND2) in the FC and 2FC groups reinforces our hypothesis regarding the enhanced energy turnover in the tissues of snails. In the present study, a significant increase in the expression of ND2 dehydrogenase in both the FC and 2FC groups was observed mainly in the digestive tract. To our knowledge, little is known regarding the regulation of ND2 expression in land snails and especially after probiotic administration. However, recent investigations have revealed a close relation between the use of probiotics and the restoring of mitochondrial function under several pathological situations [49,50].

Certain mechanisms have been proposed to clarify the benefits associated with probiotic consumption. Among them, the induction of HSP70 and HSP90, which have also been employed in gastropods, is included [51]. The obtained data in the present study underline a beneficial physiological role of probiotics on HSP expression and it seems to be tissue specific, probably related to the particular physiological properties of each tissue. Particularly, a gradual increase in the HSP70 mRNA was observed in the control group by day 90. However, the expression of the corresponding gene was faster and stronger in the FC and 2FC groups, indicating enhanced upregulation of HSP70 in the hepatopancreas of snails in the presence of probiotics. The latter is closely related to the changes in the levels of HSP70, supporting further the induction of HSPs in the presence of probiotics. The mechanism of *Lactobacillus acidophilus* probiotics in upregulating HSP70 mRNA levels still remains unknown [52]. It is established that HSP70 proteins are central components of the cellular network of molecular chaperones and folding catalysts. Specifically, they assist a large variety of protein-folding processes in the cell by transient association of their substrate binding domain with short hydrophobic peptide segments within their substrate proteins [53]. The latter enhances the hypothesis that probiotics play a crucial role in protein folding, thus indicating a strong involvement in cellular integration and homeostasis in the hepatopancreas of *C. aspersum*. Apart from HSP70, HSP90 was upregulated in the presence of probiotics in the FC and 2FC groups. It has been reported that HSP90 is involved in several cellular key processes, including cell survival, cell-cycle control, hormone signaling, oxidative stress, and apoptosis [30,54]. Nevertheless, the results obtained in the present study do not suggest a clear mechanism of HSP upregulation in the presence of probiotics, and it remains to be answered in future studies. On the other hand, however, we could suggest that the induction of HSPs, when probiotics are included in the diet, is beneficial and it might be associated with the downregulation of apoptotic pathway. In contrast to our results, other studies have found a decrease in the heat shock response under the effect of probiotic administration. Specifically, a multi-strain probiotic compound containing *Bacillus mesentericus, Bacillus coagulans, Enterococcus faecalis*, and *Clostridium butyricum* resulted in decreased HSP levels in the serum of piglets. This downregulating effect of probiotics on HSPs has been attributed to the decreased formation of free radicals, and an increase in the total antioxidant capability [55]. Similarly, the probiotic *Phaeobacter inhibens* has resulted in a decrease in HSP70, but not in the HSP90 levels in the greater amberjack (*Seriola dumerili*, Risso 1810) during the process of metamorphosis [56].

Compared to the hepatopancreas, the pattern of relative HSP70 mRNA expression indicates that it might not be a perquisite for the physiological performance of the digestive tract and mantle of snails. Compared to the control group, however, the present results revealed marked increases in HSP70 and HSP90 protein levels in these two tissues of the FC and 2FC groups. The latter strongly indicates that the presence of probiotics trigger transcriptional and translational phenomena resulting in the upregulation of both HSP70 and HSP90. Several studies have demonstrated that mRNA expression may not be related to changes in protein activity per se [57], resulting in a lower level of correlation [58] or a delayed response between mRNA and protein expression [59,60]. It has been reported that mRNA levels may increase quickly and then decrease rapidly, whereas the levels of the corresponding protein may rise slowly and persist for a much longer period than mRNA levels [59]. Rates of protein turnover, post-translational modification, alternative splicing, translational efficiency, and other processes can also act independently of transcription to alter the proteome [61]. Overall, the notably higher protein levels of HSP90 and HSP70 between the FC and 2FC groups indicate that *C. aspersum* may require different levels of produced HSPs for the maintenance of its essential vital activity in the presence of probiotics.

The data obtained in the present work are reinforced with the views of researchers from studies carried out on the aquatic gastropod *Pomacea canaliculated* (Lamarck 1819) [51]. However, these studies have attributed these increased HSP transcriptional levels to the fact that high environmental stress can affect ATP loss and damage proteins in stressed cells [62]. Nevertheless, it seems that the addition of probiotics ameliorated these heat-induced results. Therefore, dietary fortification of piglet diets with selenium and probiotics is very beneficial, making it a feasible nutritional supplement under conditions of heat stress during the summer season [63]. In aquatic invertebrates, it has been proven that probiotics’ addition triggers a thermal and metabolic stress response, substantially improving the welfare of the reared organism [64]. Studies conducted on the feeding of terrestrial productive animals—which were subjected to heat stress conditions—with an enriched diet with selenium and probiotics, demonstrated that the HSP expression was affected in both chicken [65] and piglets [63]. Specifically, relative HSP70 and HSP90 mRNA expression was significantly increased in the presence of probiotics in the hearts of broiler chickens after exposure to heat stress for 2 h and then rapidly decreased with further exposure [65]. In young piglets, significant decreases in HSP70 and HSP27 mRNA levels were observed in the liver, kidney, and spleen [63]. However, the involvement of *Lactobacillus acidophilus* probiotics in the downregulation of HSP mRNA levels still remains unknown [52].

Our results demonstrated low levels of Bax in the digestive tract and hepatopancreas of the FC and 2FC groups, accompanied by unchanged Bcl-2 levels. The decreased Bax/Bcl-2 ratio in the present study indicates the downregulation of the pro-apoptotic processes, since as it has been shown that higher levels of Bax/Bcl-2 ratios accelerate the cleavage and downstream activation of Caspase 3, ultimately leading to cell apoptosis and tissue damage [66]. The above data are strong evidence that the inclusion of probiotics in *C. aspersum* food decreases the apoptotic response in a concentration manner, since the 2FC treatment exhibited, in general, lower apoptosis compared to the FC treatment. In previous studies, the administration of the strain *L. plantarum* 16 or *Paenibacillus polymyxa* had no significant effect on the Bax/Bcl-2 ratio since the latter demonstrated the same pattern [67]. Moreover, the probiotic mixture appears to activate the cell survival signaling pathway, leading to cardiomyocyte tolerance against apoptosis in high-fed rats [68], while a beneficial diet with oral probiotic supplementation in freshwater species *Labeo rohita* (F. Hamilton, 1822) could noticeably decrease the apoptotic process in several examined tissues [69]. Although it has been shown that members of the HSP family, and specifically HSP70, are involved in the expression of anti-apoptotic proteins such as Bcl-2 [70], our results have shown that the HSP expression maintains the Bcl-2 at a constant level throughout the experimental trial. Accordingly, the Bax/Bcl-2 ratio decreased under the FC treatments, which may be indicative of a cytoprotective role of probiotics via triggering the expression of HSPs.

## 5. Conclusions

In conclusion, the obtained data in the present work underline beneficial effects of probiotics in the diet of snails. Although the RNA/DNA ratio did not indicate any effect of probiotic supplementation on the body growth rate of the snails, the increased ND2 expression levels in the hepatopancreas are strong evidence of enhanced mitochondrial oxidation. The latter may be associated with a higher metabolic rate and hence energy turnover in the presence of probiotics. On the other hand, the decreased Bax/Bcl-2 ratio indicates downregulation of apoptotic cell death under the treatment with *L. plantarum*-enriched diets and it seems to be closely related with the observed stimulation of HSP expression. Overall, the presence of probiotics in the diet of snails probably highlights the stimulation of signaling mechanisms, enabling the animal to face potentially stressful biotic and abiotic conditions. These results will stimulate further studies at the molecular, biochemical, and genetic levels to better evaluate the benefits of probiotics in the diet of terrestrial gastropods. The latter may assist farmers to further improve the efficiency of snail farming.

## Figures and Tables

**Figure 1 animals-14-00857-f001:**
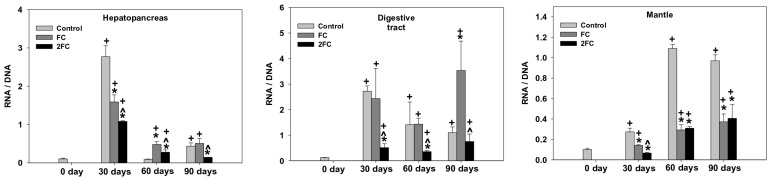
Effect of 1.25 mg (FC) and 2.5 mg (2FC) concentrations of *Lactobacillus plantarum* probiotic for 30, 60, and 90 days on the RNA/DNA ratio in the hepatopancreas, digestive tract, and mantle of *Cornu aspersum*. Values are means ± S.D.; n = 4 preparations for each group, each preparation represents 5 pooled animals. Cross (+) stands for *p* < 0.05 compared to day 0, asterisk (*) stands for *p* < 0.05 compared to control, while caret (^) stands for *p* < 0.05 of the FC group compared to the 2FC group.

**Figure 2 animals-14-00857-f002:**
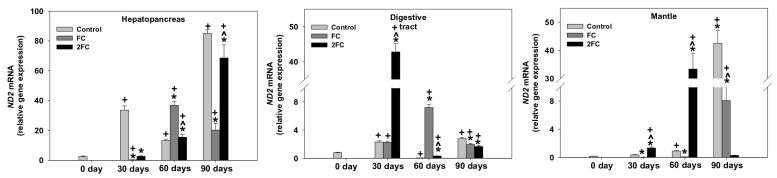
Effect of 1.25 mg (FC) and 2.5 mg (2FC) concentrations of *Lactobacillus plantarum* probiotic for 30, 60, and 90 days on the ND2 gene expression levels in the hepatopancreas, digestive tract, and mantle of *Cornu aspersum*. Values are means ± S.D.; n = 4 preparations for each group, each preparation represents 5 pooled animals. Cross (+) stands for *p* < 0.05 compared to day 0, asterisk (*) stands for *p* < 0.05 compared to control, while caret (^) stands for *p* < 0.05 of the FC group compared to the 2FC group.

**Figure 3 animals-14-00857-f003:**
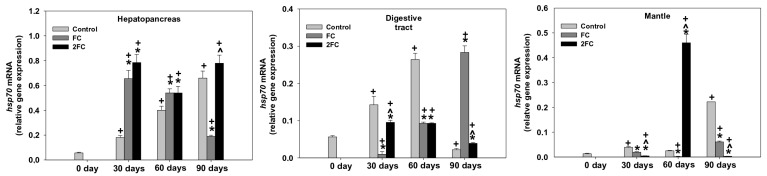
Effect of 1.25 mg (FC) and 2.5 mg (2FC) concentrations of *Lactobacillus plantarum* probiotic for 30, 60, and 90 days on the HSP70 gene expression levels in the hepatopancreas, digestive tract, and mantle of *Cornu aspersum*. Values are means ± S.D.; n = 4 preparations for each group, each preparation represents 5 pooled animals. Cross (+) stands for *p* < 0.05 compared to day 0, asterisk (*) stands for *p* < 0.05 compared to control, while caret (^) stands for *p* < 0.05 of the FC group compared to the 2FC group.

**Figure 4 animals-14-00857-f004:**
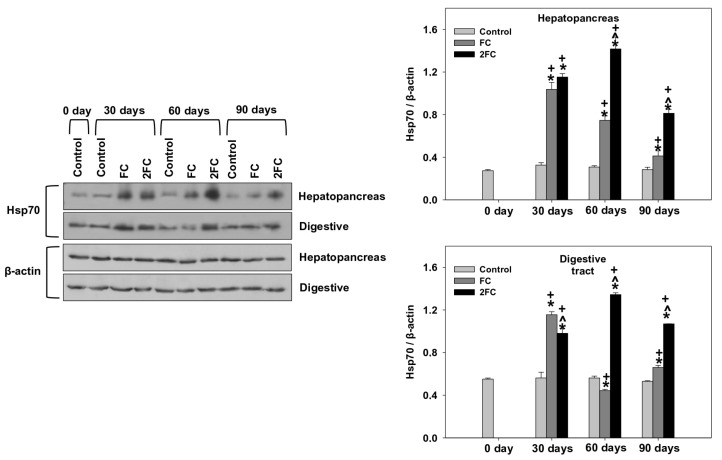
Effect of 1.25 mg (FC) and 2.5 mg (2FC) concentrations of *Lactobacillus plantarum* probiotic for 30, 60, and 90 days on relative HSP70 (HSP70/β-actin) levels in the hepatopancreas and digestive tract of *Cornu aspersum*. The quantitative histograms show the changes in the above-mentioned indicators normalized with β-actin. Representative immunoblots are shown (original blots can be found in Appendix A). Values are means ± S.D.; n = 4 preparations for each group, each preparation represents 5 pooled animals. Cross (+) stands for *p* < 0.05 compared to day 0, asterisk (*) stands for *p* < 0.05 compared to control, while caret (^) stands for *p* < 0.05 of the FC group compared to the 2FC group.

**Figure 5 animals-14-00857-f005:**
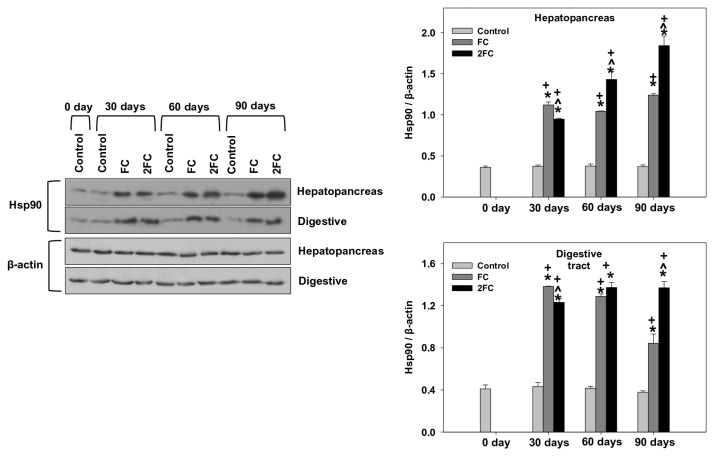
Effect of 1.25 mg (FC) and 2.5 mg (2FC) concentrations of *Lactobacillus plantarum* probiotic for 30, 60, and 90 days on relative HSP90 (HSP90/β-actin) levels in the hepatopancreas and digestive tract of *Cornu aspersum*. The quantitative histograms show the changes in the above-mentioned indicators normalized with β-actin. Representative immunoblots are shown (original blots can be found in Appendix A). Values are means ± S.D.; n = 4 preparations for each group, each preparation represents 5 pooled animals. Cross (+) stands for *p* < 0.05 compared to day 0, asterisk (*) stands for *p* < 0.05 compared to control, while caret (^) stands for *p* < 0.05 of the FC group compared to the 2FC group.

**Figure 6 animals-14-00857-f006:**
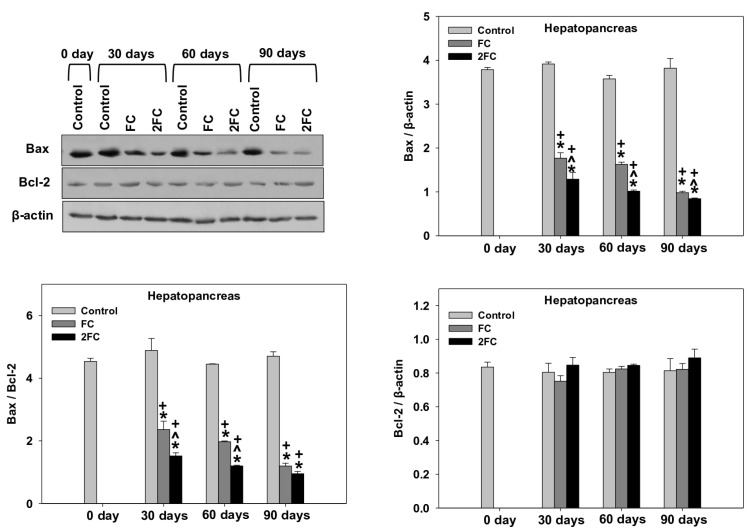
Effect of 1.25 mg (FC) and 2.5 mg (2FC) concentrations of *Lactobacillus plantarum* probiotic for 30, 60, and 90 days on Bax/Bcl-2 ratio, relative Bax (Bax/β-actin) levels and relative Bcl-2 (Bcl-2/β-actin) levels in the hepatopancreas of *Cornu aspersum*. The quantitative histograms show the changes in the above-mentioned indicators normalized with β-actin. Representative immunoblots are shown (original blots can be found in Appendix A). Values are means ± S.D.; n = 4 preparations for each group, each preparation represents 5 pooled animals. Cross (+) stands for *p* < 0.05 compared to day 0, asterisk (*) stands for *p* < 0.05 compared to control, while caret (^) stands for *p* < 0.05 of the FC group compared to the 2FC group.

**Figure 7 animals-14-00857-f007:**
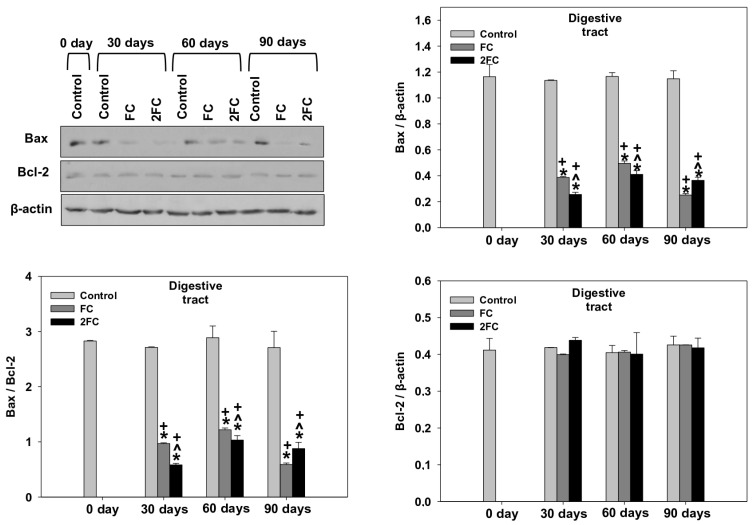
Effect of 1.25 mg (FC) and 2.5 mg (2FC) concentrations of *Lactobacillus plantarum* probiotic for 30, 60, and 90 days on Bax/Bcl-2 ratio, relative Bax (Bax/β-actin) levels and relative Bcl-2 (Bcl-2/β-actin) levels in the digestive tract of *Cornu aspersum*. The quantitative histograms show the changes in the above-mentioned indicators normalized with β-actin. Representative immunoblots are shown (original blots can be found in Appendix A). Values are means ± S.D.; n = 4 preparations for each group, each preparation represents 5 pooled animals. Cross (+) stands for *p* < 0.05 compared to day 0, asterisk (*) stands for *p* < 0.05 compared to control, while caret (^) stands for *p* < 0.05 of the FC group compared to the 2FC group.

**Table 1 animals-14-00857-t001:** Composition and nutrient levels of the ration used.

Chemical Composition (in %)			
Total nitrogenous substances	20%	Calcium (Ca)	1.00%
Total fats	3.40%	Phosphorus (P)	0.80%
Total fibrous substances	3.30%	Sodium (Na)	0.20%
Total ash	5.70%	Lysine	1.20%
Humidity	13.00%	Methionine	0.50%
Additives (per Kg of forage)			
ΒΙΤ.A: Retinol (Ε672)	12,000 mg	BIT.H: Biotin	0.20 mg
ΒΙΤ.D_3_:Cholecalciferol (Ε671)	3000 mg	Cholinechloride	500 mg
ΒΙΤ.Ε: Tocopherol acetate (Ε307)	50 mg	Co: Cobalt (CoCO_3_)(E3)	0.25 mg
ΒΙΤ.Κ3: Menadione	2.50 mg	I: Iodine [Ca(IO_3_)_2_](E2)	1.50 mg
ΒΙΤ.Β_1_: Thiamine	2 mg	Se: Selenium (Na_2_SeO_3_)(E8)	0.20 mg
ΒΙΤ.Β_2_: Riboflavin (Ε101)	4 mg	Fe: Iron (FeSO_4_)(E1)	120 mg
ΒΙΤ.Β_6_: Pyridoxine(3α831)	3 mg	Mn: Manganese(MnO)(E5)	50 mg
ΒΙΤ.Β_12_: Cyanocobalamin	0.02 mg	Cu: Copper (CuSO_4_·5H_2_O)(E4)	20 mg
Niacin	20 mg	Zn: Zinc (ZnO)(E6)	120 mg
Pantothenic acid	11 mg	Antioxidant: Ethoxycine(Ε324)	2.50 mg
ΒΙΤ.C: Folic acid	2 mg		
Composition			
Corn, soybean flour *, soybean oil, calcium carbonate, monocalcium phosphate, premix of vitamins and trace elements, sodium chloride (salt), sodium bicarbonate (soda).			

* Soybean flour comes from genetically modified soybeans, the import and processing of which is allowed freely in the European Union.

**Table 2 animals-14-00857-t002:** Primer nucleotide sequences for HSP70 and NADH genes.

Target Gene	Forward Primer (5′-3′) Reverse Primer (5′-3′)	Primer Efficiency
*HSP70*	5′-CAGCTTGAGGGCTACGTCTT-3′ 5′-GTTGCTGTGTTACTTTACCG-3′	1.96
*ND2*	5′-TGCAGAGGGAGAATCTGAGT-3′ 5′-TGGATCGAAATTATCCCGACG-3′	1.98

**Table 3 animals-14-00857-t003:** *Cornu aspersum* % mortality of the control, for FC and 2FC groups at 30, 60, and 90 days.

	30 Days	60 Days	90 Days
Control	0.26	0.33	0.3
FC	0.26	0.3	0.3
2FC	0.3	0.33	0.36

## Data Availability

Available after a reasonable request from the corresponding authors.

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
