# Peer review of "Evidence for Beneficial Physiological Responses of the Land Snail Cornu aspersum to Probiotics’ (Lactobacillus plantarum) Dietary Intervention"

_animals, 2024, doi:10.3390/ani14060857_

Round 1

Reviewer 1 Report

Comments and Suggestions for Authors

The manuscript contains many methodological errors. Additionally, the results do not fully support the conclusions. The manuscript should be carefully and thoroughly rewritten, taking into account the comments contained in the attached file.

Comments on the Quality of English Language

The English language needs to be improved, particularly concerning punctuation, grammatical errors, and articles. Some corrections have already been made to the Abstract and Simple summary. Pay attention to the clarity of the sentences

Author Response

Responses to Reviewer 1

The manuscript contains many methodological errors. Additionally, the results do not fully support the conclusions. The manuscript should be carefully and thoroughly rewritten, taking into account the comments contained in the attached file.

The English language needs to be improved, particularly concerning punctuation, grammatical errors, and articles. Some corrections have already been made to the Abstract and Simple summary. Pay attention to the clarity of the sentences.

Response: We would like to thank the reviewer for her/his constructive comments. All these comments have been thoroughly addressed in the revised manuscript. We therefore trust that the manuscript has been substantially reorganized, focused and improved.

Simple summary: Nutrition is one of the most important physiological processes for the growth and welfare of farmed animals, while interactions between the intestinal microflora and the host systems provide a key mechanism for enhancing and maintaining homeostasis. Probiotics exert a positive impact on organisms’ growth and immune response, so the administration of rations enriched with probiotics is recommended. Although the physiological role of probiotics on vertebrates’ growth and stress responses has been extensively studied in vertebrates, little is known regarding their effects on economically important invertebrates such as land snails. The aim of the present study was to investigate the effects of probiotic-enriched rations on the physiological responses in several tissues of the economically important farmed terrestrial snail, C. aspersum. Despite the absence of a direct effect on growth rate, an intense physiological response by various biomarkers was observed in all examined tissues.

Response: We thank the reviewer for the corrections made in the Simple summary. All these have been now incorporated into the revised manuscript (please see the Simple summary in the revised manuscript).

Abstract: A large variety of microorganisms ingested with food constitute animals’ intestinal microflora, enhancing and maintaining the homeostasis of the host. Ration enriched with probiotics is a method recommended to prevent undesirable conditions. To date, research has been limited to farm animals and reared fish, creating a knowledge gap concerning the effect of probiotics on the growth rate, physiological responses, and energy metabolism of invertebrates such as the land snail Cornu aspersum. Herein, juvenile snails were fed probiotic-enriched rations in two different proportions, and their growth rate was monitored over a period of three months. Additionally, RNA/DNA and Bax/Bcl-2 ratios, Hsps gene expression and their respective protein levels, as well as ND2 expression, were measured in the hepatopancreas, digestive tract, and mantle. Although snails’ growth rate was not affected, the RNA/DNA ratio presented an increase in various tissues, indicating an intense physiological response. Also, probiotic administration demonstrated low levels of the Bax/Bcl-2 ratio both in the digestive tract and hepatopancreas. Hsps’ levels were higher in the presence of probiotics, probably signalling an attempt by the animal to face potentially stressful situations. Finally, ND2 expression levels in the hepatopancreas indicate intense metabolic and antioxidant activity.

Response: We thank the reviewer for the corrections made in the Abstract. All these have been now incorporated into the revised manuscript (please see the Abstract in the revised manuscript).

Lines 51-52: Revise

Response: This sentence is omitted in the revised manuscript in order to avoid any misunderstandings to the readers.

Line 65: activity of..... ? cell, organism e.t.c.

Response: The sentence was corrected, according to the reviewer’s comment: Organisms, different fish and shellfish species (please see Introduction in the revised manuscript).

Lines 67-71: Redraft this paragraph; it's unreadable.

Response: The paragraph was rewritten as correctly recommended by the reviewer (please see the third paragraph of the Introduction in the revised manuscript).

Lines 83-85: add citation, RNA:DNA ratio is an indicator of the protein-synthesizing potential of a cell.

Response: Three citations were added and this part was rephrased in accordance to the reviewer’s comment (please see the fourth paragraph of the Introduction in the revised manuscript).

Lines 111-112: I don't see any justification for this approach. Did these snails also undergo the same process of preparation for breeding as acclimatisation? Morover, the authors should present mortality data as a function of the duration of the experiment.

Response: This sentence is omitted in the revised manuscript in order to avoid any misunderstandings to the readers. Moreover, mortality data are now presented in the revised manuscript (please see Table 3).

Line 120: What does it mean?

Response: This was deleted to avoid misunderstandings in the revised manuscript (please see section 2.1.2 in the revised manuscript)

Line 160: Why was this gene chosen as the reference? Please justify this in the discussion.

Response: This gene was used based on previous studies of gene expression in snails. This explanation was added in section 2.2.3 in the revised manuscript.

Lines 173-175: How long and under what conditions?

Response: Additional information is now provided in the revised manuscript (please see section 2.2.4 in the revised manuscript).

Lines 176-181: Really, that's how this experiment was done? The methods should be described thoroughly; the readers should not guess. The authors used antibodies targeted against mouse proteins or tested on organisms other than snails. In this case, detailed information on the antibodies, such as the amount of incubation time, etc., should be provided. In addition, authors should show antibody validation results. Verify the catalogue numbers of antibodies.

Response: Additional information is now provided in the revised manuscript in order for the SDS -protocol to be thoroughly described and clear to the readers. We also need to highlight that by mistake, other catalogue numbers than the ones of the employed antibodies were written in the submitted manuscript. However, this is now corrected in the revised manuscript and the correct catalogue numbers are provided. The antibodies used in the present study are cross-reactive against several different species including several invertebrate species which belong to evolutionary lower classes as explained in the datasheets provided for each antibody.

Specifically:

-        polyclonal rabbit anti-bcl2 (2872, Cell Signaling, Beverly, MA, USA). Species Cross-Reactivity Key: H-Human, M-Mouse, R-Rat, Hm-Hamster, Mk-Monkey, Vir-Virus, Mi-Mink, C-Chicken, Dm-D. melanogaster, X-Xenopus, Z-Zebrafish, B-Bovine, Dg-Dog, Pg-Pig, Sc-S. cerevisiae, Ce-C. elegans, Hr-Horse, GP-Guinea Pig, Rab-Rabbit, All-All Species Expected

-        polyclonal rabbit anti-bax (2772, Cell Signaling, Beverly, MA, USA). Species Cross-Reactivity Key: H-Human, M-Mouse, R-Rat, Hm-Hamster, Mk-Monkey, Vir-Virus, Mi-Mink, C-Chicken, Dm-D. melanogaster, X-Xenopus, Z-Zebrafish, B-Bovine, Dg-Dog, Pg-Pig, Sc-S. cerevisiae, Ce-C. elegans, Hr-Horse, GP-Guinea Pig, Rab-Rabbit, All-All Species Expected

-        monoclonal mouse anti-HSP70 (H5147, Sigma, Darmstadt, Germany). The antibody recognizes brain HSP70 of bovine, human, rat, rabbit, chicken, and guinea pig. It also recognizes HSP70 in Drosophila cell extract, nematode and plant as well as human fibroblast cell extract.

-        monoclonal mouse anti-HSP90 (H1775, Sigma, Darmstadt, Germany). The antibody is reactive with both the constitutive and the inducible HSP90. However, it does not bind to the native form of HSP90. Cross-reactivity has been observed with human, rabbit, rat, mice, chicken, insect (Sf9 cell line), water mold (Achlya) and wheat germ.

-        actin (anti-β actin 3700, Cell Signaling, Beverly, MA, USA). Species Cross-Reactivity Key: H-Human, M-Mouse, R-Rat, Hm-Hamster, Mk-Monkey, Vir-Virus, Mi-Mink, C-Chicken, Dm-D. melanogaster, X-Xenopus, Z-Zebrafish, B-Bovine, Dg-Dog, Pg-Pig, Sc-S. cerevisiae, Ce-C. elegans, Hr-Horse, GP-Guinea Pig, Rab-Rabbit, All-All Species Expected

Moreover, before any application of this kind, we test the antibodies used in our laboratory in order to be sure of their application. Additionally, we have to state that all results concerning western blot analysis are now expressed and normalized against actin, and the representative immunoblots are now provided (please see figures 4-7).

Line 193: The authors should conduct a microbiological analysis of the gastrointestinal content and faeces in terms of the amount of probiotic bacteria.

Response: The suggestion of the reviewer is meaningful, however the scope of the present study was to illustrate some physiological responses of snails after the inclusion of dietary supplementation of a general probiotic bacterium. It should be also noted that there are already previous studies that examined the microbiological content of gastrointestinal tissues and faeces in snails. The latter is reported in the beginning of the discussion of the revised manuscript.

Lines 197-198: Please show data about mortality over time.

Response: A new table (Table 3 in the revised manuscript) was added with mortality data in the different groups over time as recommended by the reviewer.

Line 235: To draw conclusions from the expression profile of one gene is abuse. The authors should show changes in the expression profile of other genes associated with mitochondrial metabolism, function and physiological function. The results and studies should be consistent. Why was the ND2 expression not analysed at the protein level?

Line 274: What about Hsp90 gene expression? Please provide results.

Response: It should be noted that the genome of C. aspersum has not been read yet completely and hence there are no available sequences for numerous genes. Moreover, our study represented only a first step towards understanding, by examining several molecular and biochemical indicators, the beneficial or not physiological effects of probiotics in the land snails and this topic needs further investigation.

Line 298: lack of HSP70 and HSP90 in mantle of Cornu aspersum, why?

Hsp70 and Hsp90 antibodies did not work on the mantle tissue, probably due to the high levels of mucus. Thus in order to avoid artifacts, we chose not to include mantle Hsp70 and Hsp90 results. We have employed a comparative intratissue analysis and western blot, and results were drawn only for the digestive and hepatopancreas, since no signal was produced by the mantle samples.

Line 329: Transcriptomic data should be shown. Moreover, data regarding CYT-C and caspases should be provided.

Response: So far only a few genes have been characterized in this species and thus we only chose to analyze a few biomarkers as a first step to evaluate the physiological response of the snail to probiotics dietary supplementation. Also, the two primer pairs used are described for the first time in snails.

Figure 6: I assume that the level in all groups at time 0 was the same, but this should be shown in the results. In all figures. Use relative protein instead of arbitrary units. Moreover, please normalize to total protein levels and relate to control. In all figures

Response: We thank the reviewer for the above comment. However, we must state that day 0 was the initial day before the formation of groups. We did the first sampling and immediately after it, the snails were divided into groups and the application of the feed with probiotics (FC and 2FC) was applied from that time on. Therefore, the data set for all groups is the same.

Line 353: In the discussion, the authors should refer to the analyses carried out; if we refer to analyses of other genes and proteins, it is only in relation to their results (cellular signalling, transcription factors, etc.).

Response: We thank the reviewer for the above comment. The data obtained in the present study are correlated to the existing literature in order to support our findings.

Lines 466-468: Not validated in this study.

Lines 472-474: Not validated in this study.

Line 475: There is no justification for these outcomes in the discussion.

Response: These parts of the discussion regarding the Bax/Bcl-2 ratio were rewritten as correctly mentioned by the reviewer.

Reviewer 2 Report

Comments and Suggestions for Authors

In general, this is a well written and organized manuscript. The authors included L. plantarum in diet of land snails and evaluate the levels of biomarkers in regarding to growth, energy metabolism and stress responses at RNA and protein levels. The result may provide better understanding in related area. However, there are some questions awaiting to be answered as listed below. If the authors can provide sufficient justification of these questions, I will recommend to accept this manuscript for publication.

1.     Line 120, about the feeding dosage of probiotics, how do you determine the dosages of L plantarum as 1.25 and 2.5 mg? Do you have literature evidence or natural physiological concentration of probiotics for land snail?

2.     Line 158, since the qPCR relative expression levels of genes of interest were normalized by DCt method, it is necessary to have >95% PCR efficiency to assume the amplification rate equals to two-fold per PCR cycle. Can you provide and include the PCR efficiency of all genes utilized in this experiment?

3.     For the quantitative analyses, do you pool tissue? If not, how do you generalize the bias between individuals? In addition, although it was presented n = 5 for all PCR expression profiles (legends of figure 2 and figure 3), there is no detail information to tell how do you conduct technical and biological replicates. Please provide and include this part of information in the Materials and Methods section.

Author Response

Responses to Reviewer 2

In general, this is a well written and organized manuscript. The authors included L. plantarum in diet of land snails and evaluate the levels of biomarkers in regarding to growth, energy metabolism and stress responses at RNA and protein levels. The result may provide better understanding in related area. However, there are some questions awaiting to be answered as listed below. If the authors can provide sufficient justification of these questions, I will recommend to accept this manuscript for publication.

Response: We are grateful to the reviewer for recognizing the concept and the value of our work. We hope that after addressing the comments, for each of which a response is provided below, the revise version would satisfy the editor’s and reviewers’ expectations.

  1. Line 120, about the feeding dosage of probiotics, how do you determine the dosages of L plantarum as 1.25 and 2.5 mg? Do you have literature evidence or natural physiological concentration of probiotics for land snail?

Response: The question of the reviewer is reasonable. Indeed these dosages were selected based on previous investigations of probiotic properties of other bacteria taxa in various snails. This explanation was added in section 2.1.2 in the revised manuscript, following the reviewer’s comment.

  1. Line 158, since the qPCR relative expression levels of genes of interest were normalized by DCt method, it is necessary to have >95% PCR efficiency to assume the amplification rate equals to two-fold per PCR cycle. Can you provide and include the PCR efficiency of all genes utilized in this experiment?

Response: In accordance to the reviewer’s comment, a column was added in Table 2 indicating the efficiency of both primers, which was always greater than 95% (please see Table 2 in the revised manuscript).

  1. For the quantitative analyses, do you pool tissue? If not, how do you generalize the bias between individuals? In addition, although it was presented n = 5 for all PCR expression profiles (legends of figure 2 and figure 3), there is no detail information to tell how do you conduct technical and biological replicates. Please provide and include this part of information in the Materials and Methods section.

Response: Indeed, all quantitative analyses were performed in replicates of four, with each replicate consisting of five pooled individuals (please see section 2.1.3 in the revised manuscript).

Reviewer 3 Report

Comments and Suggestions for Authors

General comments:

This study investigated the effects of probiotics enriched rations on the physiological responses in several tissues of the economically important farmed terrestrial snail Cornu aspersum. The results revealed that RNA/DNA ratio as well as the Hsp70 and ND2 expression indicated a general trend due to probiotics administration.

The manuscript could be considered for publication after being minor revisions. Some information needs to be provided as in the attachment manuscript.

Author Response

Responses to Reviewer 3

This study investigated the effects of probiotics enriched rations on the physiological responses in several tissues of the economically important farmed terrestrial snail Cornu aspersum. The results revealed that RNA/DNA ratio as well as the Hsp70 and ND2 expression indicated a general trend due to probiotics administration.

The manuscript could be considered for publication after being minor revisions. Some information needs to be provided as in the attachment manuscript.

Response: We would like to thank the reviewer for her/his constructive comments and positive criticism. All these comments have been thoroughly addressed in the revised manuscript. We therefore trust that the manuscript has been substantially reorganized, focused and improved.

Line 21: Full spelling should be provided at the first appearance.

Response: Corrected in the revised manuscript (please see revised Simple summary).

Line 29: Body weight and detailed probiotic dose information should be added in the abstract.

Response: Corrected in the revised manuscript (please see revised Abstract).

Line 64: It could be formatted as [16-18]

Response: Corrected throughout the revised manuscript.

Line 103: Here should provide approval information by Animal Welfare and Bioethical Committee?

Response: We understand the point raised by the reviewer, however, snails as invertebrates are not involved in animal welfare legislation. This has been added for clarification in the revised section, at the end of the manuscript, where ethical approval should be mentioned.

Line 104: The initial mean body weight should be provided here.

Response: In accordance to the reviewer’s comment, the initial body weight was added in the revised section 2.1.1

Line 106: There should be standard format. The same is as below.

Response: As recommended by the reviewer, the temperature format was corrected throughout the revised manuscript.

Line 192: Full spelling should be added here.

Response: Standard deviation in the revised manuscript (please see revised section 2.5).

Line 274: hsp70 or Hsp70, and italics used or not, should be uniform in the whole context.

Response: Italics were used for all gene names, whereas regular font was used for proteins. This has been checked in the whole manuscript, in accordance to the reviewer’s comment.

Line 309: S.D or SD?

Response: In accordance to the reviewer’s comment, S.D. was kept in the whole manuscript.

Line 357: 35-41

Response: Corrected in the revised manuscript.

Line 378: should be 1.5 except?

Response: Corrected in the revised manuscript.

Round 2

Reviewer 1 Report

Comments and Suggestions for Authors

Revision, 

The publication still needs improvements before publication.

1. I agree that the genome of C. aspersum has not been sequenced completely, however, mitochondrion, complete genome was known (10.1371/journal.pone.0067299).  Therefore, conclusion based on the study of one gene are highly speculative, and the authors should show changes in the expression of other mitochondrial genes, especially cox1 or 

“the increased ND2 expression levels in the hepatopancreas is strong evidence of enhanced mitochondrial oxidation”.

2.    Again the authors do not responds, Use relative protein instead of arbitrary units. Moreover, please normalize to total protein levels and relate to control. In all figures, The authors should show SDS-PAGE, not only the whole WB, in supplementary files.

3.    This sentence should be rewritten (now, it does not make sense):

       “When the protein front was extracted from the gels, these were electrophoretically transferred for 60 min.”

4.    It is hard for me to believe that using modern methods of protein isolation, purification, and precipitation cannot effectively remove excess mucus. It seems like there must be a better solution out there. Furthermore, this should be discussed in the publication to explain the lack of these results to readers; maybe include mantle Hsp70 and Hsp90 results in supplementary materials with comments. 

“Hsp70 and Hsp90 antibodies did not work on the mantle tissue, probably due to the high levels of mucus. Thus in order to avoid artifacts, we chose not to include mantle Hsp70 and Hsp90 results. We have employed a comparative intratissue analysis and western blot, and results were drawn only for the digestive and hepatopancreas, since no signal was produced by the mantle samples”

5.  Check quotation numbering (Bouétard et al. [32]).

6. The authors should discuss more closely the changes in the level of HSP gene expression in the context of the body's response to stress. Also taking into account studies showing that the addition of probiotics will reduce the expression of these genes in contrast to the results shown by the authors DOI:10.20944/preprints202309.0766.v1doi.org/10.3390/ani13132154

 7. The main question is , why HSPs increased. Heat shock protein 90 is a highly conserved molecular chaperone that assists in the maturation of many proteins involved in cellular signal transduction. As a regulator of cellular signaling processes, it is vital for the maintenance of cellular proteostasis and adaptation to environmental stresses.  So, how are probiotics supposed to regulate protein folding? I agree that Hsp protein are involved in proteins folding but probiotics? The authors should carefully reword some of the paragraphs in the discussion.

 “The latter enhances the hypothesis that probiotics play a crucial role in the protein folding, thus indicating a strong involvement in the cellular integration and   homeostasis in the hepatopancreas of C. aspersum”.  

8.  The authors should discuss more thoroughly the causes of the increase in HSP90 expression. Oxidative stress result in increased HSP90, upregulation of HSP90 is hypothesized to delay or prevent cells from entering the apoptotic pathway, Hsp90 is upregulated in stress conditions. So, If we observe HSP90 upregulation, are probiotics beneficial or not?

       “Apart from Hsp70, probiotics caused upregulation of Hsp90 in the FC and 2FC groups, suggesting an enhanced cellular homeostasis by upregulating the expression of Hsp90”

Comments on the Quality of English Language

Minor editing of English language required

Author Response

The publication still needs improvements before publication.

Response: We trust that after address of the second round of comments, the manuscript has been further improved

  1. I agree that the genome of C. aspersumhas not been sequenced completely, however, mitochondrion, complete genome was known (10.1371/journal.pone.0067299).  Therefore, conclusion based on the study of one gene are highly speculative, and the authors should show changes in the expression of other mitochondrial genes, especially cox1 or 

“the increased ND2 expression levels in the hepatopancreas is strong evidence of enhanced mitochondrial oxidation”.

Re: Indeed, the reviewer is correct, complete mitochondrial genome is already known. Besides, based on this genome we designed the primers for ND2 gene. We further agree that one gene may be speculative. Nevertheless, the present study represents a first attempt to evaluate the physiological response of a land snail in the administration of probiotics. Thus, although indeed only a few biomarkers were investigated, they represent an initial indication. Future studies that will follow may examine more markers focusing on particular physiological aspects of snail’s physiology and characterizing more genes. In an effort to comply with the reviewer’s comment, we added a sentence in the first paragraph of the discussion mentioning this limitation and future needed research.

  1. Again the authors do not responds, Use relative protein instead of arbitrary units. Moreover, please normalize to total protein levels and relate to control. In all figures, The authors should show SDS-PAGE, not only the whole WB, in supplementary files.

Re: Based on a previous comment of the reviewer, we had included in the previous revision the actin levels corresponding to the already submitted protein results. It is indicated in the figures that protein levels are expressed in relation to actin (e.g. Hsp90 / β-actin) meaning relative protein levels. Furthemore, representative blots of actin are also included, and this is also stated in the figure legends. It is obvious from the Y axis in the figures that levels were normalized to actin. However, in order to adhere to the reviewer’s comment, we have added in the results section and the figure legend the phrase “relative protein levels”.

Concerning the comment of the reviewer regarding the normalization to total protein, we must admit that by mistake we have not previously responded to the reviewer, and we apologize for that. As the reviewer is well acquainted, western blot normalization is performed either with a "housekeeping protein" such as β-actin, β-tubulin, or GAPDH which is used as a loading control, or with total protein measurement. Typically, in our laboratory we normalize our data with β-actin, a very common technique widely used by researchers worldwide. To name a few please see:

https://doi.org/10.3390/ani14020183

https://doi.org/10.3390/ani13233618

https://doi.org/10.3390/ani13182851

https://doi.org/10.3390/antiox11091840

https://doi.org/10.3390/ani13132154

https://doi.org/10.3390/ani13152537

https://doi.org/10.1016/j.bioactmat.2023.10.003

https://doi.org/10.1111/nep.14237

https://doi.org/10.1080/15384047.2023.2200705

https://doi.org/10.1002/tox.23906

https://doi.org/10.1111/bph.16202

https://doi.org/10.1016/j.numecd.2018.06.005

https://doi.org/10.1016/j.jtherbio.2022.103207

  1. This sentence should be rewritten (now, it does not make sense):

       “When the protein front was extracted from the gels, these were electrophoretically transferred for 60 min.”

Re: The sentence is now rewritten in order to be clearer and more precise.

  1. It is hard for me to believe that using modern methods of protein isolation, purification, and precipitation cannot effectively remove excess mucus. It seems like there must be a better solution out there. Furthermore, this should be discussed in the publication to explain the lack of these results to readers; maybe include mantle Hsp70 and Hsp90 results in supplementary materials with comments. 

“Hsp70 and Hsp90 antibodies did not work on the mantle tissue, probably due to the high levels of mucus. Thus in order to avoid artifacts, we chose not to include mantle Hsp70 and Hsp90 results. We have employed a comparative intratissue analysis and western blot, and results were drawn only for the digestive and hepatopancreas, since no signal was produced by the mantle samples”

Re: indeed we have tried several ways to isolate whole protein from mantle tissues. These include mechanical isolation, sonication, differential centrifugation, and even lyophilization. However, probably due to the anatomy and the nature of mantle tissue, the mucus was still present, and therefore, western-blot results regarding this tissue were not reliable. According to the reviewer’s comment we have added in the ”Results” a section explaining the absence of SDS-PAGE results in the mantle tissue.

  1. Check quotation numbering (Bouétard et al. [32]).

Re: Corrected in the revised manuscript.

  1. The authors should discuss more closely the changes in the level of HSP gene expression in the context of the body's response to stress. Also taking into account studies showing that the addition of probiotics will reduce the expression of these genes in contrast to the results shown by the authors DOI:10.20944/preprints202309.0766.v1; doi.org/10.3390/ani13132154

Re: Following the reviewer’s comment, we added an extensive part in the discussion providing a reasonable explanation for this discrepancy, that is related with the decreased formation of free radicals, and an increase in the total antioxidant capability.

  1. The main question is, why HSPs increased. Heat shock protein 90 is a highly conserved molecular chaperone that assists in the maturation of many proteins involved in cellular signal transduction. As a regulator of cellular signaling processes, it is vital for the maintenance of cellular proteostasis and adaptation to environmental stresses.  So, how are probiotics supposed to regulate protein folding? I agree that Hsp protein are involved in proteins folding but probiotics? The authors should carefully reword some of the paragraphs in the discussion.

 “The latter enhances the hypothesis that probiotics play a crucial role in the protein folding, thus indicating a strong involvement in the cellular integration and   homeostasis in the hepatopancreas of C. aspersum”.  

Re: As already mentioned in a previous response, an extensive part was added in the discussion in an effort to relate HSP protein levels increase with probiotics.

  1. The authors should discuss more thoroughly the causes of the increase in HSP90 expression. Oxidative stress result in increased HSP90, upregulation of HSP90 is hypothesized to delay or prevent cells from entering the apoptotic pathway, Hsp90 is upregulated in stress conditions. So, If we observe HSP90 upregulation, are probiotics beneficial or not?

      “Apart from Hsp70, probiotics caused upregulation of Hsp90 in the FC and 2FC groups, suggesting an enhanced cellular homeostasis by upregulating the expression of Hsp90”

Re: We believe that this comment is the most important one but in the same time difficult to approach completely. Particularly, there is an extensive part in the end of the discussion attempting to approach the general impact of probiotics that mentions the general positive potential, specifically during summer months based on available literature as well.

Reviewer 2 Report

Comments and Suggestions for Authors

The authors adequately answered  my questions and revised the manuscript accordingly, so I suggest accept it for publication.

Author Response

We thank the reviewer for the positive evaluation